# Moving towards a person-centred HIV care cascade: An exploration of potential biases and errors in routine data in South Africa

**David Etoori**[1]*, **Alison Wringe**[2], **Georges Reniers**[2,3], **Francesc Xavier Gomez-Olive**[3], **Brian Rice**[2,4]

**1** University College London, London, United Kingdom, **2** London School of Hygiene and Tropical Medicine, London, United Kingdom, **3** MRC/WITS Rural Public Health and Health Transitions Research Unit (Agincourt), School of Public Health, University of Witwatersrand, Johannesburg, South Africa, **4** University of Sheffield, School of Health and Related Research, Sheffield, United Kingdom

* d.etoori@ucl.ac.uk

**Data Availability Statement:** Data are available on reasonable request. Data may be obtained from a third party and are not publicly available. Under our data agreements with MRC/Wits Rural Public

## Abstract

In 2022, in recognition of lags in data infrastructure, the World Health Organisation (WHO) recommended the use of routinely linked individual patient data to monitor HIV programmes. The WHO also recommended a move to person-centred care to better reflect the experiences of people living with HIV. The switch from aggregated service level data to person-centred data will likely introduce some biases and errors. However, little is understood about the direction and magnitude of these biases. We investigated HIV-testing and HIV-care engagement from 2014 to 2018 in the Agincourt Health and Demographic Surveillance System (HDSS). We digitised and linked HIV patient clinic records to HDSS population data in order to estimate biases in routine clinical data. Using this linked data, we followed all individuals linked to HIV-related clinic data throughout their care pathway. We built sequences to represent these pathways. We performed sequence and cluster analyses for all individuals to categorise patterns of care engagement and identified factors associated with different engagement patterns using multinomial logistic regression. Our analyses included 4947 individuals who were linked to 5084 different patient records. We found that routine data would have inflated patient numbers by 2% due to double counting. We also found that 2% of individuals included in our analyses had received multiple HIV tests. These phenomena were driven by undocumented transfers. Further analysis of engagement patterns found a low level of stable engagement in care (<33%). Engagement fell into three distinct clusters: (i) characterised by high rates of late ART initiation, unstable engagement in care, and high mortality (53.9%), (ii) characterised by early ART initiation followed by prolonged periods of LTFU (13.7%), and (iii) characterised by early ART initiation followed by stable engagement in care (32.4%). Compared to cluster (i) older individuals were less likely to be in cluster (ii) and more likely to be in cluster (iii). Those who initiated ART prior to 2016 were more likely to be in cluster (ii) and (iii) compared to cluster (i). Those who initiated ART for PMTCT (RRR: 1.88 (95% CI: 1.45, 2.44)) or TB coinfection (RRR: 2.11 (95% CI: 1.27, 3.50)) were more likely to be in cluster (ii) when compared to those who initiated ART due to CD4 eligibility criteria. Males (RRR: 0.63 (95% CI: 0.51, 0.77)) were less likely to be in cluster (iii)

Health and Health Transitions Research Unit (Agincourt), we are unable to share the data used for these analyses directly with others. Data are however available by request here: https://www.agincourt.co.za/?page_id=188 and with permission of the MRC/Wits Rural Public Health and Health Transitions Research Unit (Agincourt).

**Funding:** This study was made possible with support from the Economic and Social Research Council (ES/JS00021/1 to ED), the Bill and Melinda Gates Foundation for the MeSH Consortium (OPP1120138 to BR), the Bill and Melinda Gates Foundation ALPHA grant (OPP1164897 to GR), and the MRC SHAPE UTT grant (MR/P014313/1 to AW). The funders had no role in study design, data collection and analysis, decision to publish, or preparation of the manuscript.

**Competing interests:** The authors have declared that no competing interests exist.

compared to cluster (i) as were those who initiated ART for PMTCT (RRR: 0.77 (95% CI: 0.62, 0.97)) or under test and treat guidelines when compared to those who initiated ART due to CD4 eligibility. Only a minority of patients are consistently engaged in care while the majority cycle between engagement and disengagement. Individual level data could be useful in monitoring programmes and accurately reporting patient figures if it is of high quality, has minimal missingness and is properly linked in order to account for biases that accrue from using this kind of data.

## Introduction

With antiretroviral therapy (ART) programmes entering their third decade in sub-Saharan Africa (SSA), there is now a concerted effort to attain sustainable HIV epidemic control by 2030 [1–3]. Since their inception in the early 2000s, SSA ART programmes have evolved rapidly and have done a commendable job in expanding access to treatment for people living with HIV (PLHIV) [4, 5]. Expansion has been characterised by ART availability in a greater number of facilities, decentralised service delivery to bring ART closer to patients, and patient-centred care with multiple models of treatment delivery, all leading to improved convenience for patients [6]. In South Africa, the benefits of treatment are reflected in the finding that the life expectancy of PLHIV who are virally suppressed is only slightly lower than among people who are HIV-uninfected. However, PLHIV who are not virally suppressed have substantially lower life expectancy that uninfected individuals [7].

The expansion in treatment programmes has not been matched by an appropriate growth in data infrastructure, with many programmes in SSA still reliant on paper registers. Where electronic medical records (EMR) have been implemented, they are often standalone, rather than being networked across health facilities, resulting in programmes often only being able to monitor facility-based retention rather than whole programme retention [8, 9]. This limitation in monitoring individual care pathways is of concern given the importance of measuring and promoting optimal adherence to ART to achieve viral suppression, and the need to better understand patterns of engagement with HIV services, as well as transfers between these services.

In recognition that as the number of PLHIV in care increases so too does the number of care pathways, researchers have proposed a person-centred cascade that takes what some have called the "side door" into consideration [10]. This new cascade is designed to better reflect individual pathways of care including, for example, cases of loss to follow-up where individuals have reengaged at a different facility. An ability to monitor pathways of care such as this would ultimately show that ART programmes are doing better than has previously been reported [10, 11]. However, the current data infrastructure does not have the capability to do this.

In 2022, the World Health Organisation (WHO) published guidelines on strategic information for HIV advocating for a shift away from aggregated service level data to a people-centred approach supported by disaggregated longitudinal routine data [12, 13]. These guidelines included advice on patient monitoring systems, recommending the use of unique identifiers and promoting the routine collection of patient-level data for reporting on programme, national and global indicators.

In SSA advances are being made in moving towards a patient-centred approach. For example, in South Africa, efforts are underway to consolidate person-centred clinical data across government services [14]. Record linkage using demographic surveillance data has also shown

promise in improving reporting of patient outcomes in the national treatment database [15]. The ongoing transfer of paper-based clinical registers to electronic reporting, however, introduces new biases and errors. While the general direction of the potential biases has been recognised [15], there have been few efforts to ascertain the magnitude of these biases and how they may affect global targets. It is important to fill this knowledge gap as inaccurate estimates not only lead to inaccurate programme indicators but may lead to insufficient resource allocation [16], and bias projections for packages such as SPECTRUM due to erroneous modelling parameters [17].

In this study, we use public health facility ART data linked to population-based data in rural South Africa to evaluate this routine data source (health facility data) for the potential biases that might accrue when utilising it to measure programmatic indicators (e.g., numbers tested, new HIV diagnoses, numbers engaged in care) and progress towards national and global targets.

## Methods

### Setting and study population

Data for this study are taken from the Agincourt Health and Demographic Surveillance System (HDSS) run by the MRC/Wits Rural Public Health and Health Transitions Research Unit (Agincourt). Agincourt comprises 120,000 residents in 31 villages and is located in Mpumalanga province, South Africa where HIV prevalence was 22.8% among adults 15 to 49 years in 2017 [18–20]. Since 1992, the MRC/Wits Agincourt Research Unit has conducted an annual demographic surveillance survey where vital events (births, deaths and migration data) are collected from residents, based on a comprehensive household registration system [19, 21–23]. Since 2014, HIV patient visits to ART clinics in the area have been logged by fieldworkers and linked to the HDSS using Point-of-Contact Interactive Record Linkage (PIRL) which has been described in detail elsewhere [24, 25]. The HDSS also collects verbal autopsy data to ascertain probable causes of death [26, 27].

The study population included all ART patients aged 18 years or older, enrolled at an ART clinic (8 clinics in total) after record linkage was established in April 2014 and who had been matched to a resident record within the HDSS database. Test and treat was adopted in 2016 [28]. This cohort was followed until July 11, 2018, when data extraction occurred. De-identified clinical and demographic data used in these analyses were accessed on August 1, 2019.

### Data preparation and statistical analysis

As facilities usually organise refill schedules around a 28-day schedule (2 days of extra pills), each refill was equated to one month (1 month equals 28 days). As we had no adherence data, we assumed that individuals were taking treatment as prescribed. For each eligible individual, we allocated a treatment pathway of up to 56 months (maximum follow-up in the data) from the first date that an ART facility record was opened for them (either HIV diagnosis or ART initiation). Considering the data requirements of a person-centred cascade, we assigned care pathways into seven possible states (Table 1).

We identified clusters with similar care engagement patterns using Optimal Matching distances [29, 30]. Using a binary variable coded as 1 if an individual was a member of a given cluster and 0 otherwise, factors associated with membership to each engagement-cluster were determined using multinomial logistic regression. We calculated counts and proportions for socio-demographic and baseline clinical characteristics and report confidence intervals for proportions of patients with multiple HIV tests. For continuous variables, we report the median and interquartile range. Missingness in exposure variables ranged from 4–7%. In

**Table 1. Description of possible states.**

| State | Definition |
|---|---|
| HIV+ no ART | Either diagnosed HIV-positive but ART not initiated at one of the facilities under surveillance, or ART initiated but with no follow-up visits following initiation. |
| On ART | At least one health facility visit following the ART initiation visit. Individuals remained on ART until the next scheduled visit. |
| Late | Up to 3 months late for a scheduled ART refill visit. |
| Lost to follow-up (LTFU) | More than 3 months late for a scheduled ART refill visit. |
| Deceased | Registered as deceased from data extracted from HDSS records. |
| Transferred | Treatment collection from a health facility within the HDSS other than the initiating facility. |
| Re-engaged | Resumed treatment at initiating facility following LTFU (i.e., returned following a period of >3 months after a scheduled ART refill visit). |

logistic regression models, we only conducted complete case analyses. Sequence analyses were conducted using the *TraMineR* R-package [31]. All other analyses were conducted using Stata 16 [32]. To explore potential biases and errors we considered individuals who have tested more than once, those who have been linked to more than one clinic record, and those LTFU.

## Ethical approval

Ethical approval was obtained from the London School of Hygiene and Tropical Medicine Research Ethics Committee, the University of Witwatersrand Human Research Ethics Committee, and the Mpumalanga Department of Health Public Health Research Committee. All methods were carried out in accordance with relevant guidelines and regulations and written informed consent was obtained from all participants and from a parent and/or legal guardian of minors (subjects under 18) in the study.

## Results

### Population characteristics

Over the study period, 8079 patient records were entered into the PIRL database. Of these, 5084 (62.9%) were linked to an HDSS resident and met the inclusion criteria. These 5084 patient records were linked to 4947 unique HDSS residents (134 unique individuals linked to multiple patient records). (S1 Fig) The 4947 individuals had a median age of 34 years (IQR: 27, 43) at ART initiation, and median baseline CD4 of 272 cells/μL (IQR: 140, 435). The 134 individuals with multiple linkages had a median age of 28 years (IQR: 24, 35), a median baseline CD4 of 268 cells/μL (IQR: 149, 425), 114 (85.1%) were female, 54 (40.3%) initiated ART in 2015, 33 (24.6%) in 2016, and 31 (23.1%) in 2014; and 63 (47.0%) initiated ART due to CD4 criteria and 38 (28.4%) for prevention of mother-to-child transmission (PMTCT). Of the 134 (2.7%) with multiple patient records, 127 (2.6%) had transferred once, 3 (0.1%) had transferred twice, and 4 (0.1%) had re-engaged at the same facility but had been given a new file number. Of 133 total transfers, 106 (79.7%) were undocumented according to PIRL records.

Based on patient records, 226 (4.6%) patients were recorded as transfers-in from facilities outside the HDSS and 47 (0.9%) were recorded as transfers-in from facilities within the HDSS at their first facility. For the 47 transfers-in from HDSS facilities, we could not identify any earlier records within PIRL, suggesting missed links or that they never had a treatment file opened at these facilities (e.g., newly diagnosed who request to start treatment at another facility).

## Testing and ART initiation statistics

Of 4947 individuals, 93 [1.9% (95% CI: 1.5–2.3)] had multiple HIV tests conducted. The overwhelming majority of patients with multiple HIV tests were among those who had transferred once (i.e., linked to two records) [90/127, 70.9% (95% CI: 62.1–78.6)] and the remainder were for those who had transferred twice (i.e., linked to three records) [3/3, 100% (95% CI: 29.2–100)]. Multiple HIV tests were more common for undocumented transfers (89/106 [84.0% (95% CI: 75.6–90.4)] compared to documented transfers 4/27 [14.8% (95% CI: 4.2–33.7)]). Median time between tests was 438.5 days (IQR: 222–805).

Almost half of the 4947 individuals (46.0%; 2277) did not initiate ART immediately (i.e., within the first month) following their diagnosis. This proportion, however, declined over the study period from 48.2% (432/897)) in 2014 to 32.7% (122/373) in 2018.

## Engagement sequences and clusters

The 4947 individuals contributed 147,651 person-months of engagement data to the sequence analysis corresponding to 85509 (57.9%) months in care, 47306 (32.0%) months out of care (LTFU), 11226 (7.6%) months late for a scheduled appointment, and 3610 (2.4%) for those deceased. Fig 1 illustrates the proportion of patients in each state over their engagement period (in months) by reason for ART initiation. Patients who initiated treatment for prevention of mother-to-child transmission (PMTCT) or under test and treat (strategy where persons diagnosed receive early treatment regardless of disease stage) contributed very little time [PMTCT: 161/25,949 months (0.62%); test and treat: 80/9,453 months (0.85%)] to the deceased state even later in their treatment course. Patients who initiated ART due to WHO stage (i.e., late-stage disease) had the largest contribution [466/6,937 months (6.72%)] to the deceased state.

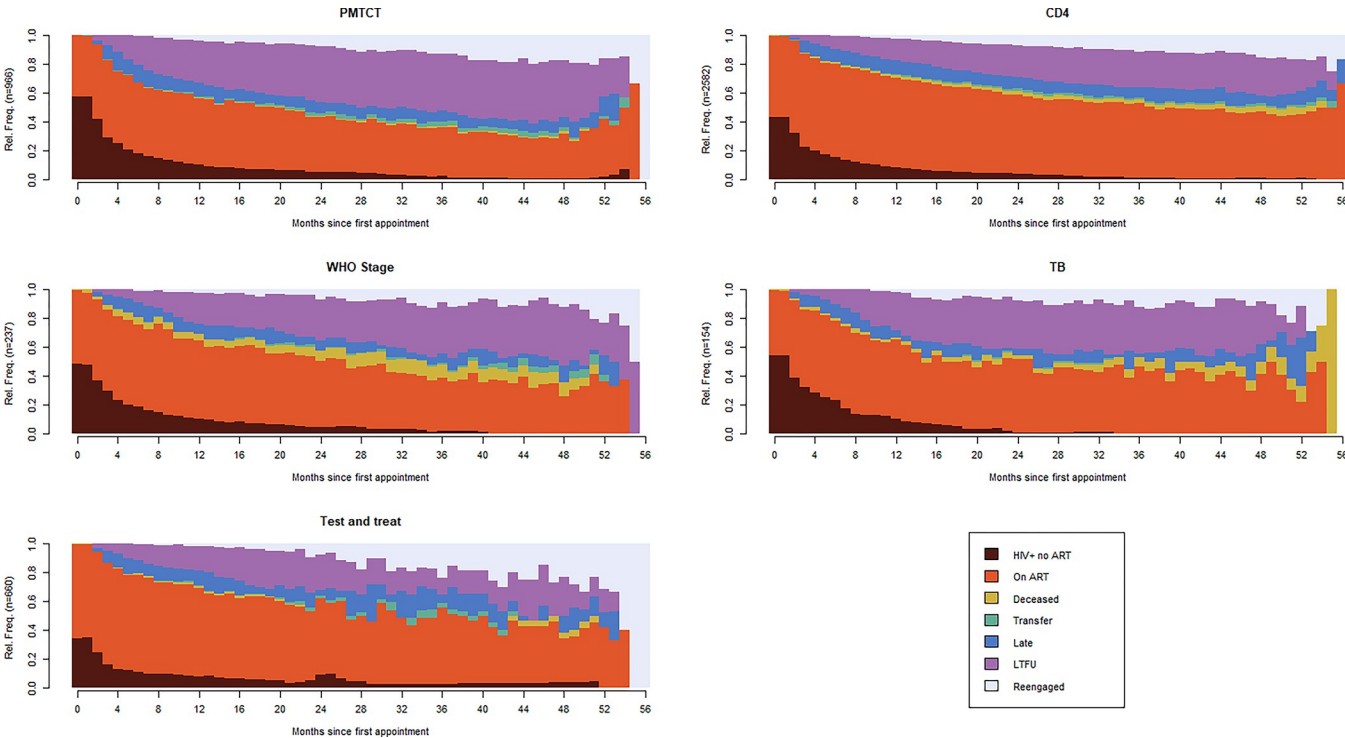

**Fig 1. Chronograms of engagement in care by reason for ART initiation.**

Some patients who initiated ART for PMTCT or under test and treat also had prolonged periods where large proportions did not initiate ART (Fig 1).

An analysis of average time spent in each state shows that males were most likely to have died (males: on average 5% of sequence time versus females: 2%) as were individuals in the 55 + age group (18–24: 1%, 25–34: 2%, 35–44: 3%, 45–54: 3%, 55+: 6%), those who initiated ART with Tuberculosis (TB), or other opportunistic infections (OIs) as measured by WHO stage at ART initiation (PMTCT: 1%, CD4: 2%, WHO stage: 6%, TB: 3%, test and treat: 1%), those who initiated ART in 2014 (2014: 4%, 2015: 2%, 2016: 2%, 2017: 1%, 2018: 1%), or with lower baseline CD4 (<100: 6%, 100–199: 3%, 200–349: 2%, 350–499: 1%, 500+: 1%). Younger patients were most likely to be LTFU (18–24: 30%, 25–34: 23%, 35–44: 17%, 45–54: 15%, 55+: 10%), as were patients who initiated ART for PMTCT (PMTCT: 29%, CD4: 17%, WHO stage: 21%, TB: 24%, test and treat: 13%), and patients who initiated ART in earlier years (2014: 20%, 2015: 22%, 2016: 20%, 2017: 12%, 2018: 2%). There were no discernible differences in person-months spent in the late sequence state. (S2–S6 Figs)

We identified three distinct patterns of engagement through cluster analysis of the 4947 sequences: (i) characterised by high rates of late ART initiation, unstable engagement in care, and high mortality (n = 2666, 53.9%), (ii) characterised by early ART initiation followed by prolonged periods of LTFU (n = 679, 13.7%), and (iii) characterised by early ART initiation followed by stable engagement in care (n = 1602, 32.4%). (Table 2) Fig 2 shows individual patient trajectories further illustrating that even within the three broader engagement patterns, patient experiences of care are not monolithic. For example, for patients in the stable engagement category, there are intermittent periods of LTFU and for patients in the prolonged LTFU category, there are periods of re-engagement in care.

In the multinomial logistic regression model, older individuals (35 and older) were less likely to be in cluster 2 (early ART/prolonged LTFU) compared to cluster 1 (late ART/unstable) [e.g., RRR 55+ years versus 25–34 years: 0.44 (95% CI: 0.28, 0.72)]. Individuals who initiated ART for PMTCT when compared to those who initiated ART due to CD4 eligibility criteria were more likely to be in cluster 2 than cluster 1 [RRR: 1.88 (95% CI: 1.45, 2.44)] as were those who initiated ART due to TB coinfection. Individuals who initiated ART prior to 2016 were more likely to be in cluster 2 compared to cluster 1 [2014 vs 2016 RRR: 1.97 (95% CI: 1.49, 2.61)]. Older individuals were more likely to be in cluster 3 (early ART/stable) compared to cluster 1 as were individuals who initiated ART prior to 2016. Males were less likely to be in cluster 3 compared to cluster 1 as were individuals who initiated ART for PMTCT and those who initiated ART under test and treat (Table 3).

## Discussion

In recognition that people often receive services, including HIV diagnoses, multiple times in the same or different locations, and that care pathways following a diagnosis can often be complex, WHO in 2022 recommended the collection and use of routinely linked individual data on HIV diagnoses and care to monitor people's movement over time between facilities and locations [33]. To better reflect the actual experiences of people living with diagnosed HIV, WHO also recommended the use of a new person-centred cascade. Ehrenkranz et al present a cascade that can track individuals as they cycle between disengagement and reengagement in care [11]. To achieve this individual-level data linked over time and space are required. Using a novel approach linking population-based demographic surveillance data and clinic records, we followed individuals across their treatment journey in order to assess the added value of disaggregated longitudinal patient-level data and identify and describe potential biases and errors.

**Table 2. Demographic and clinical characteristics of the study cohort and engagement-cluster members.**

| | Demographics | | Engagement clusters | | | | | |
|---|---|---|---|---|---|---|---|---|
| | All | % | Late ART/ Unstable | % | Early ART/Prolonged LTFU | % | Early ART/Stable | % |
| | 4947 | | 2666 | | 679 | | 1602 | |
| **Age group** | | | | | | | | |
| 18–24 | 766 | 15.5 | 431 | 16.2 | 162 | 23.9 | 173 | 10.8 |
| 25–34 | 1831 | 37 | 987 | 37.0 | 296 | 43.6 | 548 | 34.2 |
| 35–44 | 1264 | 25.5 | 687 | 25.8 | 133 | 19.6 | 444 | 27.7 |
| 45–54 | 637 | 12.9 | 341 | 12.8 | 58 | 8.5 | 238 | 14.9 |
| 55+ | 449 | 9.1 | 220 | 8.2 | 30 | 4.4 | 199 | 12.4 |
| **Baseline CD4** | | | | | | | | |
| <100 | 868 | 17.6 | 469 | 17.6 | 116 | 17.1 | 283 | 17.7 |
| 100–199 | 869 | 17.6 | 467 | 17.5 | 104 | 15.3 | 298 | 18.6 |
| 200–349 | 1244 | 25.1 | 656 | 24.6 | 184 | 27.1 | 404 | 25.2 |
| 350–499 | 882 | 17.8 | 427 | 16 | 128 | 18.9 | 327 | 20.4 |
| 500+ | 870 | 17.6 | 491 | 18.4 | 123 | 18.1 | 256 | 16.0 |
| Missing | 214 | 4.3 | 156 | 5.9 | 24 | 3.5 | 34 | 2.1 |
| **Sex** | | | | | | | | |
| Female | 3740 | 75.6 | 1943 | 72.9 | 517 | 76.1 | 1280 | 79.9 |
| Male | 1207 | 24.4 | 723 | 27.1 | 162 | 23.9 | 322 | 20.1 |
| **Baseline WHO stage** | | | | | | | | |
| I | 3572 | 72.2 | 1978 | 74.2 | 464 | 68.3 | 1130 | 70.5 |
| II | 574 | 11.6 | 264 | 9.9 | 71 | 10.5 | 239 | 14.9 |
| III | 385 | 7.8 | 195 | 7.3 | 67 | 9.9 | 123 | 7.7 |
| IV | 56 | 1.1 | 29 | 1.1 | 7 | 1.0 | 20 | 1.3 |
| Missing | 360 | 7.3 | 200 | 7.5 | 70 | 10.3 | 90 | 5.6 |
| **ART reason** | | | | | | | | |
| PMTCT | 966 | 19.5 | 494 | 18.5 | 213 | 31.4 | 259 | 16.2 |
| CD4 | 2582 | 52.2 | 1200 | 45 | 297 | 43.7 | 1085 | 67.7 |
| WHO stage | 237 | 4.8 | 117 | 4.4 | 38 | 5.6 | 82 | 5.1 |
| TB | 154 | 3.1 | 76 | 2.9 | 32 | 4.7 | 46 | 2.9 |
| Test and Treat | 660 | 13.4 | 576 | 21.6 | 26 | 3.8 | 58 | 3.6 |
| Missing | 348 | 7.0 | 203 | 7.6 | 73 | 10.8 | 72 | 4.5 |
| **ART year** | | | | | | | | |
| 2014 | 897 | 18.1 | 244 | 9.2 | 162 | 23.9 | 491 | 30.7 |
| 2015 | 1246 | 25.2 | 331 | 12.4 | 277 | 40.8 | 638 | 39.8 |
| 2016 | 1395 | 28.2 | 697 | 26.1 | 225 | 33.1 | 473 | 29.5 |
| 2017 | 1036 | 20.9 | 1021 | 38.3 | 15 | 2.2 | 0 | 0.0 |
| 2018 | 373 | 7.5 | 373 | 14.0 | 0 | 0.0 | 0 | 0.0 |

In our cohort of South African patients, we found 2% inflation of patient numbers due to double counting as a result of undocumented transfer. Of the 2.7% of people transferring between facilities during the study period, 80% of these were undocumented. We also report 2% of individuals having multiple positive tests, with individuals with undocumented transfers more likely to have multiple HIV diagnoses. HIV testing data in this setting has not been digitised and we relied heavily on tests that had been recorded in patient records. As such, this is likely an underestimation of retesting in this setting. This is especially pertinent as one study where testing data was digitised and linked to individuals in the Western Cape of South Africa found that 51% of positive diagnoses were from individuals previously diagnosed [34]. As

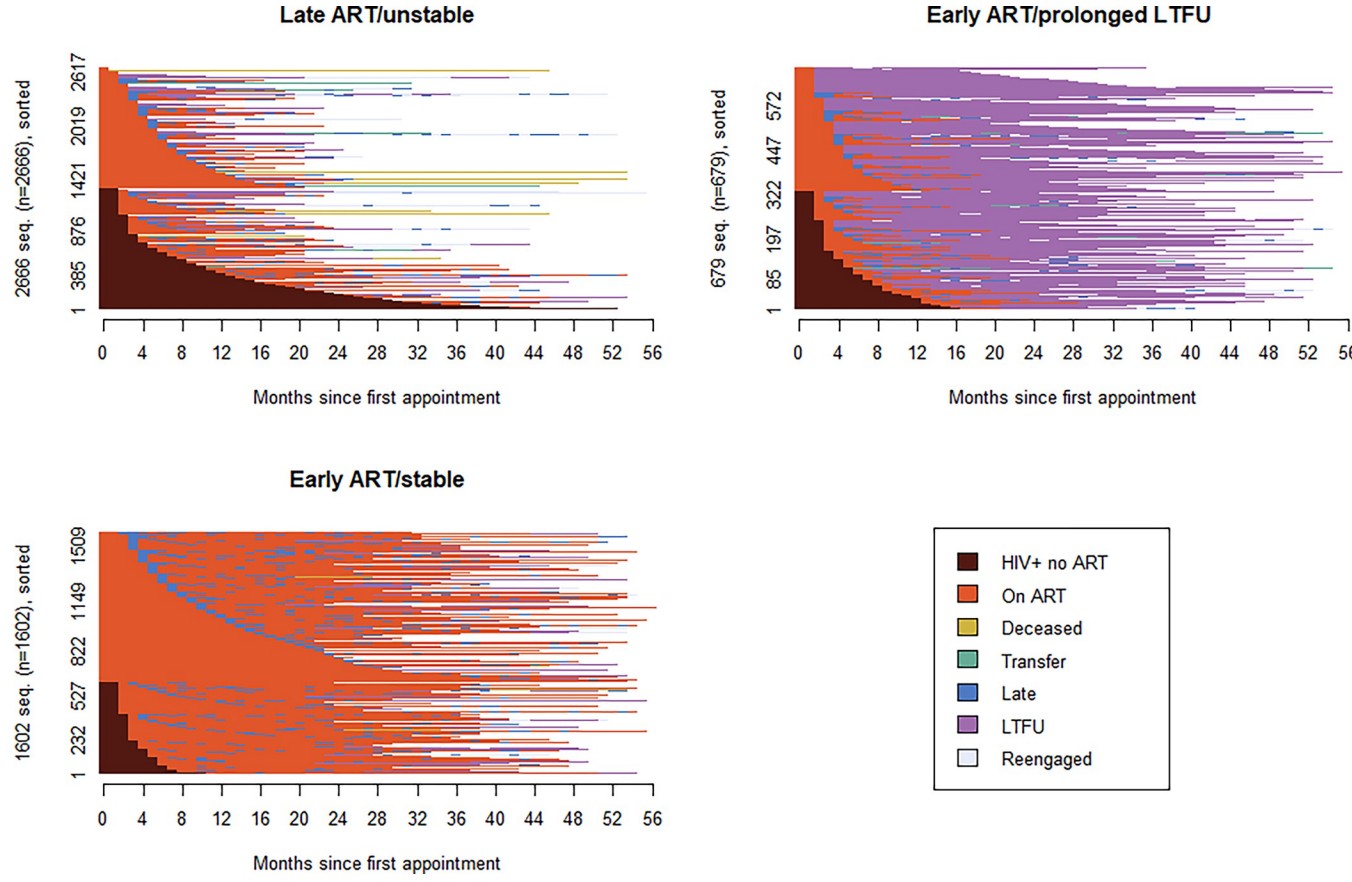

**Fig 2. Index plots of the three engagement clusters showing individual patient trajectories.**

ART is increasingly offered to healthier patients, undocumented transfers are becoming a more common occurrence in treatment programmes likely linked to labour migration [35], illustrating the need for robust cascades that takes into consideration, movements between clinics and regions and non-linear care pathways.

Using analysis of patient engagement sequences, we identified low levels (<33%) of stable engagement in care, categorising three distinct patterns of engagement. These were (1) high rates of late ART initiation, unstable engagement in care, and high mortality, (2) early ART initiation followed by prolonged periods of LTFU, and (3) early ART initiation followed by stable engagement in care. Historically, HIV programmes have used aggregate data which lacks the nuance of individual level data. The cluster analysis demonstrates the multiple ways that individuals engage with care showing that this does not follow a linear path. This is something that would not be possible with aggregate data. The clusters could also potentially assist programmes in estimating how many individuals within their programmes might have similar HIV care engagement profiles. We found that early ART initiation is still lagging, similar to other studies [36] and could reflect several reasons including individual readiness. Additionally, TB coinfection is common in this setting and delayed ART is recommended in such cases [37]. Individuals who were pregnant or who had OIs were more likely to initiate ART early consistent with treatment recommendations. However, these groups were likely to experience prolonged LTFU similar to findings from other studies [38]. Our findings suggest that two thirds of the treatment cohort have cyclical engagement patterns. We saw that LTFU is not a

**Table 3. Factors associated with engagement-cluster membership.**

| | Early ART/Prolonged LTFU (n = 679) vs Late ART/Unstable | | Early ART/Stable (n = 1,602) vs Late ART/Unstable | |
|---|---|---|---|---|
| | RRR (95% CI) | p-value | RRR (95% CI) | p-value |
| **Age group** | | | | |
| 18–24 | 1.42 (1.08, 1.87) | 0.012 | 0.84 (0.65, 1.08) | 0.18 |
| 25–34 | Reference | __ | Reference | __ |
| 35–44 | 0.75 (0.57, 0.99) | 0.04 | 1.29 (1.05, 1.60) | 0.016 |
| 45–54 | 0.63 (0.43, 0.92) | 0.017 | 1.29 (0.99, 1.68) | 0.057 |
| 55+ | 0.44 (0.28, 0.72) | 0.001 | 1.38 (1.03, 1.86) | 0.03 |
| **Sex** | | | | |
| Female | Reference | __ | Reference | __ |
| Male | 1.23 (0.95, 1.59) | 0.123 | 0.63 (0.51, 0.77) | <0.001 |
| **ART reason** | | | | |
| PMTCT | 1.88 (1.45, 2.44) | <0.001 | 0.77 (0.62, 0.97) | 0.024 |
| CD4 | Reference | __ | Reference | __ |
| WHO stage | 1.24 (0.81, 1.89) | 0.322 | 0.75 (0.53, 1.05) | 0.095 |
| TB | 2.11 (1.27, 3.50) | 0.004 | 0.87 (0.54, 1.38) | 0.55 |
| Test and Treat | 0.73 (0.47, 1.16) | 0.186 | 0.51 (0.36, 0.72) | <0.001 |
| **ART year** | | | | |
| 2014 | 1.97 (1.49, 2.61) | <0.001 | 2.95 (2.39, 3.63) | <0.001 |
| 2015 | 2.61 (2.06, 3.30) | <0.001 | 2.79 (2.31, 3.36) | <0.001 |
| 2016 | Reference | __ | Reference | __ |
| 2017 | 0.04 (0.02, 0.07) | <0.001 | Omitted | __ |
| 2018 | Omitted | __ | Omitted | __ |

terminal state, and individuals move between periods of engagement and disengagement, something that only patient-centred data can capture. We also found that engagement sequences are as unique as the individual, as such, programmes will need to evolve to offer tailored care to fit patients' individual needs.

Although we were able to link individuals across facilities, we report there being a large proportion of individuals who remain LTFU from this network. This likely speaks to two points. Firstly, the South African population is highly mobile [39], a consequence of historic inequality where economic opportunities were only available in select parts of the country. We have shown previously that individuals from this region transferred their care as far as Cape Town, in the Western Cape [35]. Secondly, ascertainment of deaths is still difficult even with demographic surveillance data. Given that membership in the prolonged LTFU category was associated with TB coinfection as the ART initiation reason (a factor associated with high mortality in other cohorts [40, 41]), it is likely that some of these individuals might have died. Our findings illustrate that there is still a need for routine reporting mechanisms that accurately document mortality a gap also recognised by WHO, which recommends cross referencing other data sources where deaths are recorded to improve accuracy of this information [33]. We have reported previously that LTFU misclassifies a large proportion of patient outcomes [15]. As such, ascertainment of outcomes following LTFU will be important in order to more accurately estimate programmatic metrics. We should note that we were unable to link 37% of patients who claimed residency in the HDSS which might have resolved some of the LTFU.

Our study has some limitations. Linkage errors, especially due to missed matches, may bias our findings. Our linkage covered a small group of public health facilities offering ART in a rural community and did not extend to several private facilities that offer services to the community or to public facilities in surrounding areas where some members of the community

could be receiving care. More extensive geographical linkage, and the inclusion of public and private facilities could reveal higher proportions of retesting and silent transfers, and as such our estimates should be interpreted as lower limits for adjustment factors for estimates from routine programme data. Due to the routine nature of data, some variables had a high level of missing values. As we did not have adherence data, the assumption that individuals were in care between treatment refills might misclassify some individuals. Optimal matching has some limitations similar to other statistical models used for cluster analysis (e.g., a statistical technique cannot speak to individual lived experiences).

In our setting, we not only highlight the feasibility and utility of linking individual level data across time and space but also report fairly modest frequency of retesting and double counting of patients on ART, highlighting the importance of silent transfers in increasing the occurrence of these phenomena. Our findings suggest that HIV programmes in SSA may need to pay particular attention to younger individuals, pregnant women, individuals with OIs, those with higher CD4 counts at diagnosis, and males as these groups are least likely to remain stable on ART and could benefit from extra interventions.

To better understand and improve programme performance, we need to strengthen data quality, sustainability and use through precise analysis and the characterisation and correction of potential biases. With the aspiration of populating provincial and national level person-centred HIV care cascades, we need to expand the reach of data linkage across facilities to more accurately describe individual level pathways of care, thereby minimising potential biases and errors.

## Supporting information

**S1 Checklist. Inclusivity in global research.**
(DOCX)

**S1 Fig. Flow chart of individuals included in the analysis.**
(DOCX)

**S2 Fig. Proportion of sequence time spent in each state on average by age.**
(DOCX)

**S3 Fig. Proportion of sequence time spent in each state on average by ART initiation reason.**
(DOCX)

**S4 Fig. Proportion of sequence time spent in each state on average by baseline CD4.**
(DOCX)

**S5 Fig. Proportion of sequence time spent in each state on average by sex.**
(DOCX)

**S6 Fig. Proportion of sequence time spent in each state on average by year of ART initiation.**
(DOCX)

## Acknowledgments

The authors would like to thank all the participants in the study.

## Author Contributions

**Conceptualization:** David Etoori, Alison Wringe, Georges Reniers.

**Data curation:** David Etoori.

**Formal analysis:** David Etoori.

**Funding acquisition:** Alison Wringe, Georges Reniers, Brian Rice.

**Investigation:** David Etoori.

**Methodology:** Francesc Xavier Gomez-Olive, Brian Rice.

**Project administration:** Francesc Xavier Gomez-Olive.

**Supervision:** Alison Wringe, Georges Reniers, Francesc Xavier Gomez-Olive.

**Visualization:** David Etoori.

**Writing – original draft:** David Etoori, Brian Rice.

**Writing – review & editing:** David Etoori, Alison Wringe, Georges Reniers, Francesc Xavier Gomez-Olive, Brian Rice.

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
