## [Decision Letter · Decision Letter 0]

28 Nov 2023

PGPH-D-23-01814

Moving towards a person-centred HIV care cascade: an exploration of potential biases and errors in routine data in South Africa

Dear Dr. Etoori,

Thank you for submitting your manuscript to PLOS Global Public Health. After careful consideration, we feel that it has merit but does not fully meet PLOS Global Public Health’s publication criteria as it currently stands. Therefore, we invite you to submit a revised version of the manuscript that addresses the points raised during the review process.

EDITOR: 

1. Kindly respond to all the comments from reviewers.

2. the primary objective of your study says 

'we use a novel approach linking public health facilities offering ART in rural north-eastern South Africa with population- based data, to evaluate routine data for the potential biases that might accrue when utilising it to measure programmatic indicators and progress towards national and global targets.'a. The novel approach that you refer to here- is this what you developed for the purpose of the study or is this something that the health system has developed to link public health facilities?b. what are the programmatic indicators that your are referring to?

3. How much of missing data was found and how was missing data handled.

4. Were there any specific assumptions made for the purpose of analysis?

4. How does identification of clusters help achieve your objective of identifying potential biases?

5. Is logistic regression and the OR the best approach to identify associated factors especially when longitudinal data was available. What was the outcome variable used in this analysis?

6. Are there any limitations wrt the analytical/ statistical methods used.

We look forward to receiving your revised manuscript.

Kind regards,

Rashmi Josephine Rodrigues, M.D., Ph.D.

Academic Editor

Journal Requirements:

2. Please provide separate figure files in .tif or .eps format only and remove any figures embedded in your manuscript file. Please also ensure all files are under our size limit of 10MB.

3. Please amend your detailed Financial Disclosure statement. This is published with the article. It must therefore be completed in full sentences and contain the exact wording you wish to be published.

Additional Editor Comments (if provided):

Reviewers' comments:

Reviewer's Responses to Questions

**Comments to the Author**

1. Does this manuscript meet PLOS Global Public Health’s publication criteria? Is the manuscript technically sound, and do the data support the conclusions? The manuscript must describe methodologically and ethically rigorous research with conclusions that are appropriately drawn based on the data presented.

Reviewer #1: Partly

Reviewer #2: Yes

2. Has the statistical analysis been performed appropriately and rigorously?

Reviewer #1: I don't know

Reviewer #2: Yes

3. Have the authors made all data underlying the findings in their manuscript fully available (please refer to the Data Availability Statement at the start of the manuscript PDF file)?

Reviewer #1: No

Reviewer #2: Yes

4. Is the manuscript presented in an intelligible fashion and written in standard English?

Reviewer #1: No

Reviewer #2: Yes

5. Review Comments to the Author

Reviewer #1: The authors address an important issue in HIV programmes - the transition from aggregated to person-centered monitoring on the HIV care cascade is important to improving outcomes. The authors are also well-positioned to address the topic because of the PIRL Database.

While the authors have access to high quality data, the manuscript could be improved by (1) being clear on terminology so that the reader is able to follow, and (2) better alignment of the problem statement/rationale with results. Regarding the latter point, there needs to be careful though into the results presented and the theme of potential biases/errors vs. broader/general descriptive analyses that do not add any answers to the research question. The lack of consistency in the use of terms makes it difficult to interpret tables and results.

Below are specific comments:

Line 30 – Please clarify – this statement “The switch to routine data will likely introduce some biases and errors.” – the switch to routine data from what?

Line 46 - a low level of sustained engagement in care 47 (<33%) – considering that significant portion of the abstract speaks to engagement patterns – it will be helpful to understand what “sustained” engagement means. Also, does this differ from “stable engagement” used in line 50.

Line 51 – Please clarify what the numbers in the brackets represent…” individuals who initiated ART in 2017 versus 2016 (77.44 (43.33, 52 138.39)), and under test and treat versus due to CD4 eligibility (1.78 (1.31, 2.42))...”

Line 54 – Is there a number missing after WHO Stage? “…with OIs [WHO stage: 1.57 (1.07, 2.30); TB: 2.47 (1.60, 55 3.80))…”

Line 56 – please clarify that 55+ is in “years”

Line 56 – Please clarify the comparison group for this statement to improve clarity “those who initiated ART due to CD4 eligibility” (i.e., compared with who?)

Line 77 – Please provide further clarity to this statement – “The expansion in treatment programmes has not been matched by an appropriate growth in data infrastructure, with many programmes still reliant on paper registers.” Please clarify where the setting to which this statement should be inferred. The authors use references 8 and 9, but the stated references do not fully support this generalization.

Line 83 – consider revising this sentence, there is repetition - and the need to better understand patterns of engagement, disengagement and engagement with HIV services.

Line 131 – Authors specify when the cohort follow up ended (This cohort was followed until July 11, 2018). Please clarify when follow up started.

Line 165 – “Of the 137 (2.8%) with multiple 166 patient records, 127 (2.6%) had transferred once, 3 (0.1%) had transferred twice, and 4 (0.1%) had re-engaged at the same facility but had been given a new file number” – should the breakdown add to 137 (i.e., 127+3+4)?

Line 167 – do the 133 total transfers refer to those with multiple patient records. If they do, then the text above shows that there were 130 transfers.

Table 2 – introduces new terms that have not been defined in Table 1 – unstable, ‘prolonged’ LTFU, early ART, stable. Because of this, it is difficult to interpret Table 2.

Line 180 – please clarify what numbers in brackets represent - [90/127, 70.9% (62.1-78.6)]

Reviewer #2: Important paper that addresses a knowledge gap in estimation of PLHIV in care, patterns and pathways of engagement/disengagement into and out of care

Study sought to explore potential biases and errors in estimation of PLHIV in care based on reconciliation or a lack thereof, in HDSS vs. PIRL – it assessed for individuals who had had multiple testing for HIV, been linked to more than one record, or LTFU

Results section

Line 159: Include a flow chart/diagram from HDSS to PIRL - showing study participants included/excluded based on criteria referred to, (un)/linked to unique identifiers and those included in the analysis

Line 165: Would be good to profile these 137 PLHIV with multiple patient records (social and clinical and demographics). Even better would be a follow-up study to understand reasons for their transfers to contextualize these findings (although this wasn’t the aim of the study).

Lines 169-170: Where are the 226 coming from? Not clear which denominator of the prior listed numbers authors are referring to and text seems fragmented from rest – would be good to make clear how it links to prior text or other section of paper.

Lines 172-173: Does this mean these patients were transferred without any documentation or is it possible that their linking identifiers were somehow not captured in the system? Or could it be that the transferring facilities chose not to issue their clinical data/files because the patients were being transferred to the new facilities as new to ART care? Do we know if these weren’t new PLHIV initiating ART or re-engaging in care after long periods of LTFU?

Lines 178-181Interesting that the majority of patients who had had multiple testing had transferred once. Does this mean that there was no association between multiple testing and multiple linkages?

Lines 200-210: Is it possible to present a trend test p-value for each of the categories to highlight the significance of change in proportion at each of the categorical levels?

Lines 237: Ages 35-54 left out?

Discussion

Line 186: How does the 32.7% statistic of not initiating same day ART in your study compare with the national statistics. Are there reasons authors think uptake of this recommendation has lagged?

The discussion is silent about figure 1 findings of patient care pathways. Aren’t these worth mentioning?

Similarly, not much is mentioned about factors associated with belonging to any of the three engagement clusters.

From the study findings are there any recommendations for HIV program managers and policy makers to consider so more PLHIV are enrolled in the Early ART/Stable engagement cluster?

6. PLOS authors have the option to publish the peer review history of their article (what does this mean?). If published, this will include your full peer review and any attached files.

**Do you want your identity to be public for this peer review?** For information about this choice, including consent withdrawal, please see our Privacy Policy.

Reviewer #1: No

Reviewer #2: **Yes: **Dr. Nelson Kalema

While revising your submission, please upload your figure files to the Preflight Analysis and Conversion Engine (PACE) digital diagnostic tool, https://pacev2.apexcovantage.com/. PACE helps ensure that figures meet PLOS requirements. To use PACE, you must first register as a user. Registration is free. Then, l

---

## [Decision Letter · Decision Letter 1]

7 Mar 2024

PGPH-D-23-01814R1

Moving towards a person-centred HIV care cascade: an exploration of potential biases and errors in routine data in South Africa

Dear Dr. Etoori,

Thank you for submitting your manuscript to PLOS Global Public Health. After careful consideration, we feel that it has merit but does not fully meet PLOS Global Public Health’s publication criteria as it currently stands. Therefore, we invite you to submit a revised version of the manuscript that addresses the points raised during the review process.

EDITOR: 

Please understand that we needed for identify a fresh reviewer for your manuscript as the previous reviewers were unavailable. Therefore, I request you to kindly consider the reviewers comments and provide clarifications necessary clarifications or explanations as appropriate.

We look forward to receiving your revised manuscript.

Kind regards,

Rashmi Josephine Rodrigues, M.D., Ph.D.

Academic Editor

Journal Requirements:

1. Tables should not be uploaded as individual files. Please remove these files and include the Tables in your manuscript file as editable, cell-based objects. For more information about how to format tables, see our guidelines:

https://journals.plos.org/globalpublichealth/s/tables

Additional Editor Comments (if provided):

Reviewers' comments:

Reviewer's Responses to Questions

**Comments to the Author**

1. If the authors have adequately addressed your comments raised in a previous round of review and you feel that this manuscript is now acceptable for publication, you may indicate that here to bypass the “Comments to the Author” section, enter your conflict of interest statement in the “Confidential to Editor” section, and submit your "Accept" recommendation.

Reviewer #1: All comments have been addressed

Reviewer #3: All comments have been addressed

2. Does this manuscript meet PLOS Global Public Health’s publication criteria? Is the manuscript technically sound, and do the data support the conclusions? The manuscript must describe methodologically and ethically rigorous research with conclusions that are appropriately drawn based on the data presented.

Reviewer #1: Yes

Reviewer #3: Yes

3. Has the statistical analysis been performed appropriately and rigorously?

Reviewer #1: Yes

Reviewer #3: Yes

4. Have the authors made all data underlying the findings in their manuscript fully available (please refer to the Data Availability Statement at the start of the manuscript PDF file)?

Reviewer #1: Yes

Reviewer #3: Yes

5. Is the manuscript presented in an intelligible fashion and written in standard English?

Reviewer #1: Yes

Reviewer #3: Yes

6. Review Comments to the Author

Reviewer #1: Thank you for addressing the comments.

Reviewer #3: The article presents valuable insights into the biases inherent within routine clinical data used for HIV program monitoring. Its exploration of patient journeys through treatment is timely, given WHO’s emphasis on person-centered data and individual engagement patterns.

1. The introduction outlines the context and challenges in HIV care data management, but the study's specific objective must be clearly stated. A concise statement of the objectives would enhance the reader's understanding.

2. Line 148: Instead of 3 logistic regression model, use a single multinomial regression model. It would improve the precision of the estimates.

3. Line 175, give full form of PMTCT at the first instance. Similarly, give full form of all abbreviations on the first instance.

4. Figure 1, the graphs do not have a title or labels on the x-axis.

5. Lines 167 – 240: To enhance the readability and interpretation of the results, consider organizing the extensive numerical data into well-structured tables.

6. In the logistic regression model for 'Late ART/Unstable' in Table 3. Could you please clarify why age wasn't included as a dependent variable? Is it because age does not have statistical significance. If so, is there any reason for that.

7. PLOS authors have the option to publish the peer review history of their article (what does this mean?). If published, this will include your full peer review and any attached files.

**Do you want your identity to be public for this peer review?** For information about this choice, including consent withdrawal, please see our Privacy Policy.

Reviewer #1: No

Reviewer #3: No

---

## [Decision Letter · Decision Letter 2]

10 May 2024

Moving towards a person-centred HIV care cascade: an exploration of potential biases and errors in routine data in South Africa

PGPH-D-23-01814R2

Dear Mr. Etoori,

We are pleased to inform you that your manuscript 'Moving towards a person-centred HIV care cascade: an exploration of potential biases and errors in routine data in South Africa' has been provisionally accepted for publication in PLOS Global Public Health.

Best regards,

Rashmi Josephine Rodrigues, M.D., Ph.D.

Academic Editor

Reviewer Comments (if any, and for reference):

Reviewer's Responses to Questions

**Comments to the Author**

1. If the authors have adequately addressed your comments raised in a previous round of review and you feel that this manuscript is now acceptable for publication, you may indicate that here to bypass the “Comments to the Author” section, enter your conflict of interest statement in the “Confidential to Editor” section, and submit your "Accept" recommendation.

Reviewer #3: All comments have been addressed

2. Does this manuscript meet PLOS Global Public Health’s publication criteria? Is the manuscript technically sound, and do the data support the conclusions? The manuscript must describe methodologically and ethically rigorous research with conclusions that are appropriately drawn based on the data presented.

Reviewer #3: Yes

3. Has the statistical analysis been performed appropriately and rigorously?

Reviewer #3: Yes

4. Have the authors made all data underlying the findings in their manuscript fully available (please refer to the Data Availability Statement at the start of the manuscript PDF file)?

Reviewer #3: No

5. Is the manuscript presented in an intelligible fashion and written in standard English?

Reviewer #3: Yes

6. Review Comments to the Author

Reviewer #3: All suggestions have been incorporated. Hence I recommend for publication.

7. PLOS authors have the option to publish the peer review history of their article (what does this mean?). If published, this will include your full peer review and any attached files.

**Do you want your identity to be public for this peer review?** For information about this choice, including consent withdrawal, please see our Privacy Policy.

Reviewer #3: **Yes: **Santu Ghosh
